# The Four Homeostasis Knights: In Balance upon Post-Translational Modifications

**DOI:** 10.3390/ijms232214480

**Published:** 2022-11-21

**Authors:** Stefania Pieroni, Marilena Castelli, Danilo Piobbico, Simona Ferracchiato, Damiano Scopetti, Nicola Di-Iacovo, Maria Agnese Della-Fazia, Giuseppe Servillo

**Affiliations:** Department of Medicine and Surgery, University of Perugia, Piazzale L. Severi 1, 06129 Perugia, Italy

**Keywords:** post-translational modifications, ubiquitin and ubiquitin-like proteins, oncogenes, onco-suppressors, RAS, MYC, RB, p53

## Abstract

A cancer outcome is a multifactorial event that comes from both exogenous injuries and an endogenous predisposing background. The healthy state is guaranteed by the fine-tuning of genes controlling cell proliferation, differentiation, and development, whose alteration induces cellular behavioral changes finally leading to cancer. The function of proteins in cells and tissues is controlled at both the transcriptional and translational level, and the mechanism allowing them to carry out their functions is not only a matter of level. A major challenge to the cell is to guarantee that proteins are made, folded, assembled and delivered to function properly, like and even more than other proteins when referring to oncogenes and onco-suppressors products. Over genetic, epigenetic, transcriptional, and translational control, protein synthesis depends on additional steps of regulation. Post-translational modifications are reversible and dynamic processes that allow the cell to rapidly modulate protein amounts and function. Among them, ubiquitination and ubiquitin-like modifications modulate the stability and control the activity of most of the proteins that manage cell cycle, immune responses, apoptosis, and senescence. The crosstalk between ubiquitination and ubiquitin-like modifications and post-translational modifications is a keystone to quickly update the activation state of many proteins responsible for the orchestration of cell metabolism. In this light, the correct activity of post-translational machinery is essential to prevent the development of cancer. Here we summarize the main post-translational modifications engaged in controlling the activity of the principal oncogenes and tumor suppressors genes involved in the development of most human cancers.

## 1. Introduction

In addition to proper lifestyle and behavior, the healthy state of individuals is maintained through the ability to counteract the external insults suffered by the human body. Biologically, this role is carried out by genes encoding for proteins that control cell proliferation, differentiation, and development. The alteration in their modulating functions impacts on specific signal transduction pathways, inducing cellular behavioral changes finally leading to cancer [1].

Genetically, two classes of genes are related to cancer development: oncogenes and tumor suppressor genes (TSGs). Proto-oncogenes are dominantly “activated” to oncogenes through alteration: in quantity, so that they become overexpressed; or in quality, so that a mutant and/or dysfunctional protein is expressed. In both cases, cells are exposed to an aberrant growth burst. As a counteraction, tumor suppressor genes are committed to restrain cells from unregulated proliferation through different pathways: mainly apoptosis, autophagy and senescence. The TSGs inactivation usually are “biallelic” since their function is recessive, and it releases cells from growth control finally provoking carcinogenesis. Since these alterations exert a pivotal role in damage sensing and in the prevention of uncontrolled growth, the cellular amount of pseudo-oncogenes and TSGs and their activation pathways must be strictly under control [2].

According to these observations, we will delineate the biological strategies engaged to control the cellular amount of such fundamental proteins, especially focusing on four of the most important oncogenes/TSGs trough the analysis of the main post-translational modifications involved.

## 2. Transcriptional and Translational Control of Protein Synthesis

The biosynthesis of proteins in cells and tissues is controlled at both the transcriptional and translational levels, but the global mechanism that allows them to carry out their functions does not exclusively involve the mere number of proteins [3].

The effectiveness of protein synthesis, particularly when talking about oncogenes and tumor suppressor genes, depends on the different steps of control.

### 2.1. Genetic and Epigenetic Control

Genetic control is affected by genomic or genic alteration, including modifications to the chromosome structures and gene sequences (i.e., genetic mutations, transposition or deletions of chromosome parts, genomic instability, loss of heterozygosity-LOH, and gene copy number variation-CNV) [4,5]. Basically, the alteration leads to the production of either normal or altered proteins both in quality and/or amount, also by impairment between the codifying and controlling sequences.

The perturbations of the epigenetic control change the heritable state of gene expression and chromatin organization, without modifying nucleotide sequences (i.e., histone modifications, DNA methylation and/or acetylation, and loss of imprinting-LOI) [6,7]. In particular, DNA methylation and histone modification rate modulate the transcription of several master genes in the control of cell growth, differentiation, apoptosis, and transformation [8,9,10,11], also by establishing inactive chromatin conformations to silence gene expression [12,13]. Methylation-associated silencing has been demonstrated in various genes, including tumor suppressor genes [14,15].

As a therapeutic approach, the induction of the re-expression of epigenetically silenced TSGs could provide a means to suppress cancer outcome and to increase responsiveness to new drugs [6].

### 2.2. Translational Control

Translational control guarantees the correct correspondence between the mRNA code and protein sequence. The mis-transduction of genetic code can drive the synthesis of altered or misfolded proteins that are unable to fulfil their specific biological role. To avoid this event, a ribosome-associated protein quality control implements a series of “protocols” to allow the biogenesis of functional proteins (i.e., folding chaperones, translational speed control, and quality control of polypeptides) [16,17].

## 3. Post-Translational Modifications

The process of protein synthesis must be integrated and balanced. On the one hand, protein degradation maintains the appropriate protein levels, and on the other, protein modification ensures their correct fold and function. The amount of each protein is a key factor to regulating cell function and metabolism, but the mere accounting of proteins is not enough. Keeping the healthy state requires not only that the right proteins are synthetized and quantitatively regulated, but also that the right protein is in the right place and time during a cell’s life. With this aim, a major challenge for the cell is to guarantee that proteins are made, folded, assembled, and delivered to function properly [18].

This check-up mechanism is a key step when talking about cancer. In the normal cell the number of TSG proteins must be finely tuned since it triggers the response to oncogene activation, or other environmental insults and stress stimuli. The living cell needs to be able to quickly modulate (up or down, on or off) the protein level and localization of oncogenes and TSGs like and even more than other proteins [19,20].

One of the most-studied mechanisms of protein modulation is driven by a growing amount of moiety acting in the process described as post-translational modification. Post-translational modifications (PTMs) are dynamic to allow the cell to rapidly counteract extracellular stimuli or insults and rapidly restore the basic condition to maintain homeostasis. In general, since most PTMs are reversible, they act as a switch from the normal to active state of the target and/or vice versa. In this light, post-translational machinery is essential in preventing the development of cancer.

Although it also occurs in oncogenes, this strategy is particularly striking in TSGs. Indeed, TSGs activation lies in the modulation of various PTM states of the TSGs themselves and their transcriptional and non-transcriptional targets and effectors for cell survival and proliferation control [21,22]

Along with single regulatory modification, in the past decade emerging and ever more solid evidence has demonstrated the coordinated action of different PTMs acting both in parallel and sequentially [23,24,25,26].

The crosstalk between the different types of PTMs is the keystone to quickly controlling and updating the activation state of many of the master proteins in the orchestration of cell metabolism, to initiate or inhibit downstream signals and to fulfil a specific biological function. The combining of a myriad of single and/or multiple spatially and timely orchestrated modifications can govern a target protein’s life span and activity, their distribution within the cell compartments and consequently the protein interaction rate, ultimately leading to the appropriate outcome [27,28].

So far, following the canonical biological approach and the newest growing web resources, we can count a few hundred kinds of modifications occurring at 15 of the 20 amino acids present in protein sequences and tens of thousands of modification sites have been identified across the whole proteome [29]. Moreover, by using the bio-informatics approach, it is possible to conduct a proteome-wide analysis of single PTMs and crosstalk between modified residues in physical proximity displaying conservation of structural features and consensus sequences [30,31,32].

### 3.1. Types of PTMs

Conceptually the PTM machinery consists of “writers”, “erasers”, and “readers”. Writers (i.e., kinases, methyltransferases, and acetyltransferases) mediate the transfer of a modifying group of different complexity (from a chemical group to a complex polypeptide chain) onto protein substrates. Erasers (i.e., phosphatases, demethylases, and deacetylases) are responsible for the reversibility of the PTMs by mediating their removal from protein substrates. Readers act between writers and erasers since they detect and decodify the signaling written on labeled target proteins leading to protein activation/inactivation or further modifications.

Basically, PTMs can be divided into two large groups: the first provides for the addiction of chemical groups—with a wide range of complexity—at specific amino acids; the second involves the formation of a covalent linkage to a monomeric or polymeric protein moiety.

Along with phosphorylation, as the mostly diffused PTM, the first group includes acetylation, methylation, glycosylation, and some others. The second mostly observed PTM, ubiquitination, is included in the second group that counts also on SUMOylation and NEDDylation. According to different substrates and context, phosphorylation can determine the activation state of enzymes and receptors, also controlling their stability, interaction, and downstream pathways [33]. Ubiquitination acts in the modulation of protein stability, and manages fundamental biological processes, such as cell cycle, immune responses, apoptosis, and cancer. Albeit structurally similar to ubiquitination, SUMOylation pathways do not induce target degradation and have a pivotal role in the preservation of genomic integrity, transcriptional regulation of gene expression, and signal transduction [21].

The mechanism of modification for the first group is a single or a few-step enzymatic reaction. The process engaged for the second group of modification is quite complex and deserves more details. The PTM machineries for ubiquitination and SUMOylation are briefly summarized in Box 1 and Box 2.

### 3.2. PTMs Crosstalk

PTMs crosstalk may occur both intra- and interprotein [34,35]. It takes place on the same residue, as well as at proximal or distal sites along the protein sequence in a sequential mechanism in which one depends on the other [31], and it is observed between the same or different types of PTMs [28,36,37,38,39].

As a result, the interaction between PTMs can be a positive (or activating) crosstalk, since it serves as a signal for another PTM occurring downstream, or acting by itself on the protein that carries out the function, and a negative (or inactivating) crosstalk, since two PTMs compete for the same amino acid (direct competition), or the first PTM controls some further modification (indirect competition) [28,37,40]. Thus, PTM perturbation may possibly induce disease development and progression, and especially induce a predisposition to carcinogenesis [41].

### 3.3. PTMs in the Control of Oncogenes and TSGs Activity

Among oncogenes and TSGs, we will focus on four of the most important and surely impactful players in controlling cell homeostasis and health (Table 1), namely:

Functional description, types of oncogenic alteration and the main cancer sites. The Cancer Genome Atlas (TCGA) was accessed and profiled through cBioPortal (http://cbioportal.org/ (accessed on 13 October 2022). Data were from the TCGA Pan-Cancer Atlas project.

-RAS proteins as molecules integrating receptor signaling along pathways that control cellular growth;-c-MYC as a transcription factor acting as a key player in the development of many human cancers;-RB, as the first tumor suppressor gene identified;-p53, as the master gene in the preservation of genetic integrity, so far called “the guardian of the genome”.

All of them are tightly and functionally connected and their activity is closely related to PTMs

## 4. RAS Proteins

Rat sarcoma (RAS) is the most frequently mutated oncogene in human cancer [42,43]. RAS proteins are a family of GTPases able to integrate signals from different inputs to modulate cellular behavior. They fundamentally act as binary molecules switches, whose off/on condition depends on GDP or GTP binding. The three mammalian RAS genes (*HRAS, NRAS*, and *KRAS*) give rise to four proteins (HRAS, NRAS, and KRAS4A/KRAS4B splice variants) because of an alternative splicing at the *KRAS* gene [44].

Box 1Ubiquitination pathway.Ubiquitin (Ub) is a 76-amino-acid, highly conserved regulatory protein firstly described by Gideon Goldstein et al. in 1975 [45,46].By covalent attachment to a target protein, Ub is able to drive it towards specific cellular pathways, including the cell cycle, apoptosis, receptor downregulation, and transcriptional control. Ub can be bound as a monomer (mono- or multiubiquitination) or other Ub may be added to a previously bound Ub (polyubiquitination). All reactions take place through the same process involving the C-terminal Gly of the binding Ub and specific residues (N-terminal methionine-M1 or lysines-K) along the target protein or the previously bound Ub. Each binding site on Ub (M1 or K6, K11, K27, K29, K33, K48, and K63) is at the basis of different linear or branched polyubiquitin chains that drives the selected target to its fate [47]. Ub can be added following a cascade of enzymatic reactions, including Ub-activating (E1), Ub-conjugating (E2), and Ub-ligating (E3) enzymes. As going from E1s towards E2s and finally E3s the substrate specificity increases together with the number of members of each group: from two E1s to ~40 E2s divided in four classes, to more than 600 E3s grouped into four different families [48,49].                    
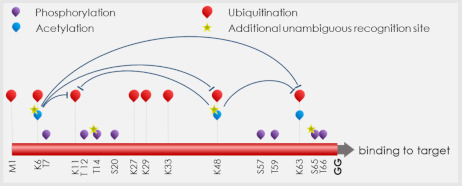
The impact of acetylation and phosphorylation on Ub chain formation. Star indicates acetylation at K6 and K48, and phosphorylation occurring at T14 and S65 representing unambiguous new recognition site for further PTMs. K11, K48, and K63 linkage are inhibited by K48 and K6 acetylation.Briefly, E1 promotes an ATP-dependent reaction that generates a thioester linkage between the C-terminus of Ub and its active Cys site. In turn, this reaction increases the Ub affinity for E2 and favors the transfer to the E2 active Cys site (trans-thioesterification) and subsequent release of the E1. E2s are characterized by a conserved catalytic domain, interacting both with E1s and E3s. E2s are key regulators of ubiquitin chain assembly, since they control both initiation and elongation of chains and allow the preassembly of Ub chains on their active sites. Some E2s are addicted to specific chain linkage, thus choosing between mono- or polyubiquitination. Moreover, by controlling different kinds of Lys linkage, they control the functionality of targets. By interacting with selected E3, they promote the recognition of specific substrate motifs [50,51,52,53].E3 ligases are extraordinarily important in determining the specific types of ubiquitinated substrate. Of the four classes of E3s, three (HECT, homologous to E6-AP C-terminus; RBR,
RING-in between-RING; and RCR, RING Cys Relay classes) were directly charged by E2 enzimes at their Cys active residue and enabled to ubiquitinate a target [54,55,56]. The fourth E3 ligases with RING (really interesting new gene) domains recruit both substrates and Ub-charged E2s. In this case, E3 ligases act as a bridge by activating E2s to transfer Ub directly from their own
active site to the substrate [57,58,59].Recent studies demonstrated that Ub itself can be subjected to phosphorylation and acetylation. Thr7, Thr12, Thr14, Thr66, Ser20, Ser57, and Tyr59 on Ub can be phosphorylated and this event modifies monoubiquitin as well as polyubiquitin chain recognition by E3s or Ub-binding proteins. Interestingly, Lys6, Lys48, and probably Lys63-acetylated Ub stabilizes the monoubiquitination state, but represses polyubiquitin chain elongation in vitro and in intact cells by affecting the noncovalent interaction of Ub with E2 enzymes. Indeed, acetylated Lys6 and Lys48 and phosphorylated Thr14 and Ser65 are unambiguously recognized by E2s, thus adding a new grade of control in the building of Ub-chains [60]. The first-identified Ub kinase PINK1 mediates phosphorylation at Ub Ser65, necessary for the ubiquitination of mitochondrial membrane proteins affecting mitophagy outcome [61,62,63].

RAS proteins are almost identical (97% homology) at their functional G-domains (aa 1–165), but at the carboxyl-terminus they differ for the hypervariable region (HVR) that mediates membrane association and directs the subcellular trafficking of RAS moieties (Figure 1). Due to its impact on cancer, the management of RAS protein activity is fundamental and, besides oncogenic mutational events and transcriptional control, lies in several PTMs that control both functional amount and localization.

### PTMs on RAS

RAS undergo a series of constitutive and highly efficient modifications that globally influence RAS activity. This processing, which occurs immediately after translation, is made of three different steps, and is collectively named CAAX processing, by the C terminus aminoacidic sequence involved (Cys-Aliphatic-Aliphatic-X, X for any aa) [64]. The outcome of CAAX processing is the enhancement and broadening of the RAS affinity spectrum with phospholipid bilayers for the intercalation into membranes. Since RAS function strongly depends on the membranes wherein it is inserted, this set of modifications is crucial for RAS signal transduction.

Conversely, phosphorylation is a conditional PTM that HRAS, NRAS, and KRAS4B have been shown to be subjected even if the functional outcome of this modification is still partially undefined. Protein kinase A (PKA), protein kinase C (PKC), and cyclic GMP-dependent protein kinases2 (PGDK2) are accounted in the number of enzymes involved in HRAS and KRAS phosphorylation [65,66]. Phosphorylation is involved in the modulation of RAS affinity to membranes. Interestingly, PKC-mediated KRAS4B phosphorylation at Ser181 affects the membrane microdomains partition by weakening the affinity to the plasma membrane. This condition promotes the translocation towards the endomembrane, including those at the mitochondria where it mediates apoptosis [67,68,69]. Moreover, at the same KRAS4B site, phosphorylation can inhibit tumor initiation acting by noncanonical wingless integrated (Wnt) signaling [70]. Non-receptor tyrosine kinase steroid receptor coactivator SRC and the ubiquitous tyrosine phosphatase SHP2 phosphorylates and dephosphorylates, respectively, RAS at Tyr32 and 64, regulating the binding to RAS effectors. In this light, phosphorylation appears as a switch enabling RAS tumorigenic activation [71,72].

Previously recognized as a major mechanism of epigenetic control, protein acetylation was firstly discovered in histones in 1964 [73] and demonstrated in non-histonic proteins in 1997 following the studies of Wei Gu on p53 [74,75,76,77]. Acetylation is observed in the wild-type and G12V KRAS-activating form in the core G-domain at multiple Lys residues (K101, K104, K128, and K147) [78]. Usually, acetylation does not strongly impact on RAS function. In particular, acetylation at K104 and K147 is reported to serve as a docking site to promote new interactions [40,79,80]. Several residues, such as K104 or 147 undergo other PTMs, as previously cited. K147 is reported also for methylation, but this modification is typical of HRAS, and is unpredictable in other small GTPases [81]. Beside the effect on the generation of different docking site for interactors, the PTMs can also act by blocking or promoting each other (acetylation, and ubiquitination), rather than altering RAS structure or intrinsic function. K5 is reported to be di-methylated and, without altering RAS conformation, it affects protein interactions. Further modifications at this position modulate RAS function [82,83]. Of note, K5 of KRAS4B undergoes polyubiquitination by the E3 ligase NEDD4-1 (neural precursor cell expressed developmentally downregulated protein 4-1) and its inhibition underlies RAS-driven tumorigenesis [84].

Box 2SUMOylation pathway.Despite the similarity of enzymatic pathways for SUMOylation and ubiquitination, there are significant distinctions between them. The first small ubiquitin-like modifier protein
(SUMO) was discovered in 1996 [85,86], but unlike Ub, five different isoforms of SUMO exist [87]. While SUMO4 and 5 functions are still largely unknown [88,89], SUMO1, 2 and 3 are better characterized. SUMO1 shares with Ub the tridimensional conformation with more than a sequence similarity (18% only). SUMO2 and 3 (collectively cited as SUMO2/3) share a 97% homology by themselves and a 50% homology with SUMO1. Unlike Ub, SUMOs require a maturation process but they show the classical C-terminus Gly-Gly end feature, enabling the link to target proteins [90]. Similarly to Ub, target conjugation occurs by a Gly-Lys isopeptide bond but unlike Ub, SUMO binding requires a typical SUMOylation consensus motif ψKxD/E (ψ being a large hydrophobic residue and x any aa) [91,92]. On the other hand, SUMO moiety binding may be mediated by weak interactions occurring at sumo interacting motifs (SIMs) that have been found in a number of proteins undergoing SUMOylation [93].As ubiquitination requires a wide repertoire of E1, E2, E3 enzymes, SUMOylation is carried on by unique E1 and E2 and a few E3s [94].SUMOylation can be mono-, poly-, or multiSUMOylation according to the SUMOs involved. Out of the three better-characterized SUMOs, only SUMO2/3 has the capacity for chain formation by recruiting Lys7, 11, 21 and Lys33 as internal SUMO attachment sites [95,96]. Specific linkage generates distinct branching patterns, as observed with Ub, with typical functional outcomes [96]. Of note, new studies found that, in addiction to mono- or multimodification, under stress conditions, SUMO1 could be tagged by SUMO2/3 to build SUMO chains [97,98].A really small number of SUMO proteases, namely sentrin-specific protease or SUMO-specific peptidase (SENPs), are engaged to reverse SUMOylation [99].SUMOylation consensus motif ψKxD/E can be post-translationally modified to widen the complexity of recognition [100], both by phosphorylation and acetylation, as for Ub. Acetylation of SUMO1 at Lys37, SUMO2 at Lys11 and Lys32, and SUMO3 at Lys33 modulates the interaction with SIMs [96,101,102]. Several non-canonical binding sites were reported [103].Given the similarities between Ub and SUMO at the mechanistic and molecular levels, it is not surprisingly that they can build together mixed chains in specific functional contexts [104].

Distinct mono-, di-, and polyubiquitin linkages have been observed at multiple lysine residues of HRAS, NRAS, and KRAS4B proteins and they can manage their function by modifying protein localization, interaction, and stability [105,106,107]. Phosphorylation can crosstalk to ubiquitination and modulate it. In response to activation by the Wnt/β-catenin signaling pathway and following Thr144 and 148 modifications by glycogen synthase kinase 3β (GSK 3β), HRAS is polyubiquitinated by β-transduction repeat-containing protein (β-phTrCP) and consequently degraded [108]. β-phTrCP and other E3s are accounted to be involved in RAS ubiquitination not only driving them to degradation. RAB5 exchange factor (Rabex-5) is a key player embedding both ubiquitin binding and E3 ligase domains for NRAS and HRAS but not KRAS modification. Rabex-5 is able to promote both mono- and di-ubiquitination and downregulates RAS function through directing non-degradational ubiquitin-mediated relocalization at the endosomes [109,110]. Beside influencing protein amount and localization, KRAS monoubiquitination was found to alter RAS interactions with regulatory proteins and effectors, thus upregulating its activity by enhancing GTP loading. In general, it was reported that, in the cell, the ubiquitinated KRAS forms are in the activated GTP-bound state and show enhanced binding to the downstream effectors as rapidly accelerated fibrosarcoma (RAF), phosphatidylinositol 3-kinase (PI3K), and RAS-like guanine nucleotide exchange factors (RalGEF) [82,83]. Regarding the modification site, Lys147 is critical for ubiquitin binding on both HRAS and KRAS and similarly for mono- and di-ubiquitination [107]. Additional sites are reported for HRAS (K170) and KRAS (K117 and K104) [82,83].

As obvious ubiquitination can be both negatively and positively modulated [111,112] and RAS mono- and di-ubiquitination can be reversed. The failure of this mechanism could be detrimental. The deubiquitinase otubain 1 (OTUB1) is found to be overexpressed in non-small cell lung carcinoma and this event can promote RAS activation and tumorigenesis [113].

In affecting the regulation of cell growth, migration, and tumorigenesis, RAS proteins are targets of SUMOylation [114,115,116]. Interestingly, Lys residues involved in SUMO binding are not the same as the ubiquitinated ones. Unpublished data by Philips and Dai describe Lys42 in HRAS, NRAS, and KRAS4B as modified upon SUMOylation by the ubiquitin-like protein SUMO3. In other studies, SUMOylation by the protein inhibitor of activated STAT (PIASγ) E3 ligase can promote MAPK signaling and tumor progression [117]. However, the physiological significance is uncertain even if downregulation of RAS signaling by Lys42 mutation results in reduced cell migration and invasion in multiple cell lines, and in tumor development in vivo [118].

## 5. MYC

The *myc* gene was discovered more than 30 years ago [119,120,121]. The proto-oncogenic version was named c-*myc* [122] and is one of the most highly deregulated oncogenes among different human cancers [123]. In mammalian cells, the three MYC variants are expressed by three independent genes encoding for the proteins: MYC or c-MYC, n-MYC (neuronal) and l-MYC (lung), being all implicated in the genesis of human malignancies [124,125,126]. MYC is a member of the MYC/MAX/MAD basic region/helix-loop-helix/leucine zipper (bHLHZ) domain transcriptional regulators, whose components act as MYC/MAX heterodimers able to bind E-box DNA consensus sequences and are repressed by MYC/MAD dimer formation (for the detailed structure refer to Figure 2) [127]. MYC aberrant activation occurs by DNA amplification, transcriptional upregulation, or protein stabilization, and when MYC levels are unbalanced, this deregulates its function [123,126,128,129]. Upon MYC deregulation cells become “egotistic”, so that they do not behave according to environmental cues and improvidently activate proliferative programs, thereby contributing to disease and cancer [130,131].

MYC is one of the most powerful transcription factors and it is thought to regulate a lot of the actively transcribed genes in the cell [132]. MYC controls fundamental biological processes, including nucleotide biosynthesis, RNA processing and protein translation, cell growth and metabolism, cell cycle progression, proliferation and differentiation, apoptosis, and senescence [133,134,135,136]. Given the role of MYC levels on cancer outcome, it is assumed that its amount and activity are under tight control and regulation during normal cell life [137,138].

### PTMs on MYC

c-MYC is a short-lived protein, and a number of PTMs are engaged in its control, over and in addition to transcriptional regulation (Figure 2).

Phosphorylation is fundamental in the control of MYC amount and function, since its proteasomal degradation is tightly dependent on its phosphorylation status. MYC phosphorylation at Thr58 (T58) and Ser62 (S62) within the transactivation domain is critical and interdependent, since S62 phosphorylation is required prior to the one at T58, and this event enables the proteasomal degradation of MYC [139]. Different signal transduction pathways, as well as cell-cycle-specific regulators manage the phosphorylation of S62, including mitogenic stimuli and RAS activation by various kinases. The phosphorylation of S62 (pS62) by RAS-RAF-MEK (Mitogen-activated protein kinase/ERK kinase)-ERK (extracellular-signal-regulated kinase) kinases cascade and/or cyclin dependent kinases (CDKs) drives MYC stabilization. pS62 condition is indispensable to allow T58 phosphorylation by GSK-3β. This path is inhibited by PI3K/Akt signaling, but is dependent on RAS pathway inactivation and results in the degradation of MYC through the ubiquitin–proteasome pathway [132,140,141].

This is a clear paradigm of the writer-reader-eraser mechanism of PTMs. As an essential controller of phosphorylation signaling, peptidyl-prolyl cis-trans isomerase NIMA-interacting 1 (PIN1), recognizes phosphorylated residues at S62 and T58 through its WW phospho-binding domain. The following isomerization of the specific phosphosite motifs (pSer/Thr-Pro) allow the intervention of PP2A dephosphorylating enzyme on S62. PP2A firstly activates GSK-3β by dephosphorylation and then, upon GSK-3β-mediated T58 phosphorylation and isomerization, recognizes the pS62/pT58 couple on MYC, catalyzes S62 dephosphorylation and enables MYC degradation [142]. On the same T58, another minor but no-less-important PTM, O-GlcNAcylation, is able to stabilize c-MYC and increase its transcriptional activity [143]. Of note, T58 falls in a hot spot frequently mutated in Burkitt’s lymphoma [144].

Several sites of phosphorylation influence the transcriptional activity of MYC by affecting MYC/MAX complex affinity. Thr358, Ser373 and Thr400 residues within the C-terminal bHLHLZ region are modified by p21/RAC1–activated kinase2 (PAK2). MYC phosphorylation at these sites decreases MYC affinity for MAX, thereby destabilizing the ternary protein–DNA complex and influencing MYC transcriptional activity [145]. Phosphopeptide mapping tools revealed additional phosphorylation sites along the MYC sequence (S71, S82, S162 or 164, S293, and S343/344) but the enzymes involved in these modifications and their functional relevance have not been defined yet [146].

cAMP response element-binding protein (CREB) binding protein (CBP)/p300, Tat interacting protein 60 (TIP60), and mammalian general control of amino-acid synthesis (mGCN5) are acetyltransferase enzymes involved in transcriptional activation upon MYC recruitment to specific promoters [147], but MYC itself is subjected to acetylation at lysine residues [148,149]. Certainly, since the occurrence of lysines in both ubiquitination and acetylation, these two modifications can potentially interfere, and acetylation at specific ubiquitination sites decreases MYC ubiquitination, enhances its stability and potentially controls the binding of interacting proteins [148,149,150]. Except for Lys417—within the leucine zipper domain—which is predicted to modulate the efficiency of MYC/MAX binding, and Lys323—within the NLS domain—which accounted for influencing MYC localization, the other Lys residues (K143, K157, K275, K317, K323, and K371) are acetylated by p300 and affect protein stability, but not the transcriptional complex formation [149,150,151,152]. MYC acetylation can be reversed by the protein deacetylase sirtuin1 (SIRT1) and generate a negative feedback loop with it, since SIRT1 is a MYC transcriptional target and once activated, interacts with and deacetylates MYC, decreasing its protein stability [153].

MYC is a short-lived protein whose half-life is around 30 min in physiological conditions. Proteasomal degradation and ubiquitination have a fundamental role in MYC turnover, however ubiquitination is found also to occur to control MYC function rather than its degradation rate [154]. MYC turnover is regulated trough the Skp1 (S-phase kinase-associated protein 1) -Cul1 (cullin1) -F-box (SCF) complex. The SCF E3 ligase complex is the largest family of E3 ligases [155]. The SCF complex is made of two parts. An invariable core with an adaptor protein Skp1, a scaffold protein Cul1 and a ring-finger protein RBX1 (RING box protein-1) or ROC1 (regulator of cullins-1). The variable component, F-box protein, confers substrate selectivity. Multiple F-boxes for MYC have been identified, thus confirming the multiple functions and the complexity of MYC activity control. SCF^Fbw7^ is the best-characterized ubiquitin E3 complex for MYC and its activity depends on S62/T58 phosphorylation cited above [156]. Other complexes depend on phosphorylation at T58, as reported for SCF^Fbw3^, involved in the cryptochrome circadian regulator 1(CRY1)-induced MYC intervention on circadian rhythms [157]. Moreover, alternative SCF complexes can antagonize MYC stabilization as reported for SCF^β-TrCP^ vs. SCF^Fbw7^. SCF^β-TrCP^ binding on MYC depends on phosphorylation at amino acids not required for Fbw7. SCF^Fbw7^ acts mainly in the S-phase and destabilizes MYC, while conversely, SCF^β-TrCP^ polyubiquitination depends on polo-like-kinase-1 (PLK1)-mediated MYC phosphorylation and take place in the G2 and M phase to modulate MYC activity by enhancing its stability [158,159]. Other SCF complexes are involved in MYC regulation in both stability and activation control. They include SCF^Skp2^ [160,161], SCF^FBXL14^ [162], SCF^FBOX28^ [163], and SCF^SPOP^ [164], with all of them involved in cell growth and tumor initiation and progression. Other identified E3s are p53-induced RING-H2 protein (PIRH2), also involved in p53 stability control [165], and HECT, UBA and WWE domain-containing E3 ubiquitin protein ligase 1 (HUWE1, also known as ARF-BP1, HectH9, MULE), engaged for activating MYC transcriptional activity by recruiting the coactivator p300 [166]. Obviously, deubiquitination plays a role in regulating MYC protein stability by modulating the ubiquitination rate. So far, six DUBs have been reported to act on MYC, including USP28, USP36, USP37, USP22, USP13, and USP7 [167].

By competing with other PTMs for the same lysine modification site, SUMOylation can interfere with protein–protein interactions and regulate protein localization, trafficking, stability and activity. At first, some studies showed that MYC is modified by SUMO at the C terminal residues Lys 323 and 326 [168,169]. Now, about ten sites are accounted to be targets for SUMOylation, and even the role of each site is often promiscuous; it is critical for MYC modulation. Moreover, deSUMOylation appears to crosstalk with ubiquitination and other PTMs to ensure the balance of MYC function. Interestingly, as a crosstalk with PTMs, PIAS1-mediated SUMOylation at Lys51 and 52 induces S62 JUN-N-terminal-kinase (JNK1)-mediated phosphorylation and thus stabilizes and activates MYC. By cooperating with the deSUMO, SEMP7, PIAS1, and ring finger protein 4 (RNF4) are able to manage the tumorigenic potential of MYC [170,171].

## 6. RB

The RB was the first tumor suppressor gene identified in the past century and was originally characterized as responsible for pediatric retinoblastoma [172]. RB exerts many cellular functions, including the control of differentiation and the regulation of apoptosis [173,174,175], quiescence, senescence [176,177,178,179], and chromosomal stability preservation [180,181,182,183,184]. The RB protein family consists of RB, and two RB-like proteins: p107 (RBL1) and p130 (RBL2). These variants share the conservation of cyclin fold pocket domains providing the docking site for binding transcription factors acting in target gene regulation [185]. The role of RB protein in controlling cell proliferation and cell cycle progression resides in its function as an interactive platform, able to allow protein interactions. PTMs have a great impact on RB scaffold features and distinct post-translational modifications, such as acetylation or phosphorylation, are capable of modulating the interaction with specific cellular partners [186,187,188,189,190] (Figure 3).

### PTMs on RB

Analysis on RB sequence and structure revealed that phosphorylation induces peculiar conformational changes in Rb, each generating discrete proteoforms underpinning the transduction into specific functional outputs. We can consider a potential code in the management of the individual site phosphorylation state, so that we can predict how the code could trigger different activities [191]. The proof of RB’s pivotal role is in the more-than-200 partners reported to interact with its platform [192]. Basically, RB acts as a repressor of cell growth and, except for a few cases, RB phosphorylation results in protein inactivation and the transcriptional derepression of genes controlling cell cycle progression [193]. RB phosphorylation is carried out mainly by CDK and checkpoint kinase 2 (CHK2) [194] and typically occurs outside the structured domains (see RB structure in Figure 3). As phosphorylation promotes conformational changes, it mediates the transition from disordered to ordered structures that mask specific docking sites for interactors, thus controlling their activation pathways [195,196,197,198]. Finally, phosphorylation promotes interprotein transition thus prohibiting the intermolecular binding to other proteins [199,200,201].

In non-cycling or quiescent cells, RB is in the hypo-phosphorylated activated form (hypo-pRB). The low rate of phosphorylation drives hypo-pRB binding to E2F transcription factors and represses DNA replication and cell cycle progression [202,203]. Conversely, when cells need to progress in the cell cycle, RB becomes hyper-phosphorylated by kinases (hyper-pRB), leading to the release of E2F factors and the transcription of S phase genes [187,204]. There are about 15 different identified phosphorylation sites and there is not a specific outcome defined for all of them. Phosphorylation inhibits the binding to different E2Fs protein isoforms (T373, S608/S612, S788/S795, and T821/T826) or to the LxCxE consensus containing proteins (T373, T821, and T826) or to other proteins (S249/T252). Some have an unknown effect (T356, and S780) or simply prime for other PTMs (S807/S811). Some sites have the double function of inducing specific factor recruitment and further RB phosphorylation, as described for Ser608/Ser612, that can activate RB phosphorylation at other sites, but is also able to induce PIN1 recruitment [205]. So far, growing data has demonstrated that the phosphorylation state is not attributable to a single enzyme but probably results from the function of a pool of cyclin/CDKs that execute specific patterns of phosphorylation, leading to a typical functional output [206]. Different kinase activation seems to reflect distinct signal transduction pathways making the pRB phosphorylation state an integration point of discrete signals controlling cell cycle progression [207].

Dephosphorylation of RB involves mainly two phosphatases PP1 and PP2A, managing the formation of distinct functional proteoforms [208]. pRB dephosphorylation occurs at the mitotic exit into G1 phase and CDKs sites undergo this event at different rates through mitotic transition. However, PP1 and PP2A activity depends on different stimuli and environmental stresses. As reported for kinase recruitment, defined signals could induce dephosphorylation at specific sites. This observation suggests a selective mechanism in stress response, rather than in normal cell cycle regulation [209,210].

From this data, we recognize the fine control of phosphorylation sites and rates as the primary mechanism of controlling RB function. In this light, p16^INK4a^ is the main controller of RB action. p16^INK4a^ is a powerful inhibitor of D-type cyclins through its binding to CDK4/6. The resulting reduction of phosphorylation rate promotes the formation of pRB/E2F complexes leading to cell cycle block at G1/S transitions [211]. In pathological conditions, it has been reported that cyclin D1 overexpression and p16^INK4a^ protein alteration induces a hyperphosphorylation of pRB and uncontrolled cell growth [212].

In addition, acetylation and methylation occur at the RB protein. As for phosphosites, the specific sites have been identified in linker sequences, specifically towards the RB carboxy-terminal domain [194]. In general, acetylation and methylation at lysine residues are induced in response to signals, such as DNA damage or cell differentiation, leading to RB activation and repression of genes. Acetylation occurs on Lys873 and 874 sites, within the cyclin-docking sequence, inhibiting kinase activity and leading to downstream hypo-phosphorylation. Further, methylation competes with acetylation resulting in hypo-phosphorylation and the recruitment of a transcriptional repressor [188,213,214,215,216]. Additionally, both acetylation and methylation are reported to control CDK activity by modifying specific lysine residue patches in the aa neighboring the linker sequences at the carboxyl terminus of RB [217].

The regulation of RB activity through the degradation pathway resides in the accurate balance between the amount of the three members of RB family, whose amount reciprocally changes during the cell cycle progression. Even if sharing the same binding pocket domain, only RB can accomplish the typical tumor suppression functions. However, the balance in the tree isoform amount ensures the timely regulated activation of cyclin/CDKs complexes engaged for RB phosphorylation and controls RB functions other than those involved in tumor suppression, including cell differentiation [218].

Even if it is not the primary means of control, the regulation of the RB amount in the cell through proteasome-mediated degradation is reported but still largely unknown. Many viral proteins are reported to physically interact with regulatory units of the 20S proteasome and control RB stability or facilitate its degradation. RB controls the expression of many key proteins involved in the cell defense to viruses, so that the RB function is a critical factor for viral replication. This strategy is supported mainly by the LxCxE motif identified by structural and biochemical studies in viral proteins interacting with RB [219,220,221,222,223,224]. The hypo-phosphorylated RB is more frequently targeted for proteasomal degradation. Both ubiquitin-dependent and independent mechanisms are described: the first at the 26S proteasome level, the second through the binding to the 20S proteasome mediated by adapter proteins as described in viral-induced degradation. Until today, only mouse double minute 2 (MDM2) is recognized as typical a E3 for RB, despite an alternative role in a 20S fashion also being described by MDM2. Moreover, acetylation occurring at Lys873 and 874 of RB results in reduced phosphorylation and increased RB affinity for MDM2 influencing its proteasomal degradation [188,225].

Ubiquitin-like SUMO proteins are engaged in the control of pRB function [226,227,228]. As for ubiquitination, SUMO preferentially targets the hypo-pRB and its function is mediated by the binding of viral and cellular inhibitors. Some papers report that SUMO targets pRB within a lysine cluster surrounding the LxCxE binding cleft, suggesting that SUMO could modulate the interaction of a pocket with distinct LxCxE-containing proteins [229] as described for ubiquitin. SUMO might modulate the formation or stability of complexes at pRB pockets, promoting an intramolecular interaction able to displace LxCxE-containing proteins, thus relieving pRB repression on E2F. A similar mechanism is hypothesized for histone deacetylase (HDACs) recruitment to transcriptional regulators [230,231]. Finally, the role of SUMO appears to be tightly related to the modulation of the RB-CDK2 interaction thus controlling the phosphorylation rate [232,233].

A critical site for SUMO binding is Lys720 tightly flanking the LxCxE cleft. Functional studies by SUMO-deficient K720R confirm an inhibitory role of SUMO on the repressive action of pRB, probably related to the displacement of a transcriptional corepressor or altered binding to E2F factors [234,235].

## 7. p53

Even if it was originally described as a tumor suppressor, up until today we recognize p53 as the more powerful controller of cellular homeostasis, acting through cell cycle arrest, apoptosis, DNA repair, cellular senescence, and other metabolic processes [236,237,238,239]. That is why currently, researchers name p53 as the “guardian of the cell” [240], rather than the “guardian of the genome” as they previously did [241].

Since p53 exerts a plethora of roles, the control of p53-mediated responses to both endogenous and exogenous stimuli must be tightly modulated in a time- and space-dependent way. Without considering the complex genetic and epigenetic control, reversible and irreversible PTMs surely represent the smartest way to dynamically modulate the wide potential of p53. p53 acts both as a monomeric and homotetrameric complex, exerting transcriptional and non-transcriptional functions, respectively [240,242]. The p53 structure is well-known and consists in a multidomain protein made of six functional domains (Figure 4). 

The characterization of each functional domain, with the modification sites within, was widely and deeply studied in past years and allowed researchers to consider PTM crosstalk as a key mechanism to balance p53 functional output [243]

As for all the short-lived proteins, the control of p53 levels is obtained mainly through a fine balance between a stably high-level of synthesis and a quickly adjustable degradation rate able to ensure the proper protein levels.

### PTMs on p53

Phosphorylation is strongly associated with protein stability and protein–protein interactions of p53 but the functional readout depends on the functional domain involved. At the steady state, some phosphorylation sites are maintained to be constitutively phosphorylated to ensure the constitutively high degradation rate of p53 (T55, S315, S362, S366, S376, and S378), since they promote the recognition by E3 ligases, such as MDM2 or β-TrCP [244,245]. Several papers demonstrated that under normal cellular growth conditions, TATA box binding protein-associated factor 1 (TAF1) stably phosphorylates p53 at Thr55, promoting the MDM2-mediated proteasomal degradation of p53. Conversely, upon DNA damage, T55 can be promptly dephosphorylated by PP2A, thus preserving p53 stability and inducing cell cycle arrest [246,247]. Similarly, the exposure to ionizing radiation leads to dephosphorylation at Ser376 (constitutively phosphorylated together with S378), which provides a docking site for 14-3-3 proteins and increases the binding affinity of p53 for its sequence-specific DNA elements [248,249].

Other phosphorylation sites are maintained un-phosphorylated to be quickly activated upon different stimuli or stresses (S6, S9, S15, T18, S20, S37, and S106) [246,247,248], since they facilitate the release of p53 from degradation. Under different stimuli, phosphorylations induced at specific sites (S15, S33, S37, S46, T81, S215, and S392) are able to manage p53 transcriptional activity [250,251,252,253,254,255,256,257,258,259,260,261]. As reported, phosphorylation at Ser33, Thr81, and Ser315 induced upon UV radiation or DNA-damaging agents provide a recognition site to activate a p53-mediated checkpoint control [262,263]. Other researchers have demonstrated the engagement of E3 ubiquitin ligase, such as Pellino1. Pellino1 is recruited to DNA damage sites when bound to p53 via pT18 and inhibits p53 degradation enabling the downstream activation of p53 responsive genes, such as p21 [264]. An inhibitor of nuclear factor kappa-B kinaseβ (IKKβ) phosphorylates p53 at Ser392 allowing the cancer cell to adapt to nutrient deprivation [261]. The same phosphorylation is also able to regulate p53 mitochondrial translocation and transcription-independent apoptosis [265]. Phosphorylation at S15 is critically involved in p53 activation towards cell growth arrest and apoptosis upon different stress exposures, such as deep hypoxia [266,267].

Acetylation occurs mainly at six specific lysine sites inside the p53 CTD (K370, K372, K373, K381, K382, and K386) in a balance between p300/CBP and several deacetylases, (i.e., histone deacetylase1 and sirtuin1) [268,269]. Globally, acetylation provides a transactivation pathway that enhances the p53 DNA-binding activity following DNA damage [74,270] and thus recruits p53 to its target gene locus to activate its functional outcome, including cell cycle arrest, senescence, or apoptosis [271]. Some studies have demonstrated that, at a steady state, p53 acetylation is fundamental for its transcriptional suppression via myeloid leukemia-associated protein SET (Suppressor of variegation, Enhancer of Zeste, Trithorax) as a key repressor of p53 activity in unstressed cells [272,273]. Since all the six C-terminal lysines can be also ubiquitylated and phosphorylated, the modification rate by phospho- or acetyl-groups is a way to modulate both p53 stability and activation.

Within DBD the acetylation status (at K101, K120, and K164) is critical for p53 target selectivity. Lys120 acetylation by TIP60 is responsible for p53 conformational changes leading to selective transcription of proapoptotic genes [274,275]. Other HAT enzymes can induce K120 acetylation resulting in function spanning from apoptosis [276] to anti-proliferative activity and senescence [277,278]. Certainly, K120 is often mutated in cancers highlighting its role in p53 modulation [274]. The other two lysines on DBD (K164 and K101) can be both acetylated by CBP/p300 [272] leading, respectively, to the induction of cell cycle arrest through p21 expression [279], and ferroptosis [273]. Finally, at the N-terminus Lys320 acetylation by p300–CBP associated factor (PCAF) prevents the phosphorylation of p53 controlling pro-survival genes fundamental to its function [272,280].

The interpretation of acetylation code is made by the cell through reader proteins. Acetyl-lysine residue readers are proteins displaying the ability to recognize specific modified sites to generate the appropriate readout. The first domain identified is the bromodomain consisting of a hydrophobic docking domain able to selectively bind acetylated lysine residues [281,282]. The CBP bromodomain specifically reads acetylated Lys382 of p53, which is indispensable for the activation of p53-induced cell cycle arrest after UV exposure [283]. Similarly, the TAF1 subunit of the transcription initiation factor TATA-box binding protein (TFIID) recognizes acetyl groups at Lys373 and 382 of p53 upon UV treatment.

As reported for acetylation, methylation was firstly studied in histones. In p53 functional control, methylation is observed firstly in lysines K370, K372, K373, and K382 having both positive and negative effects on p53 stability and transcriptional activity. While SET7/9-promoted methylation at K372 stabilizes p53 and enhances some target gene transcription (i.e., p21), K382 and K370 methylation induced by SET8 and SET and MYND (Myeloid-Nervy-DEAF1) domain-containing protein 2 (SMYD2), respectively, reduces p53-regulated transcription [284,285,286]. K370 and K373 can also undergo demethylation leading to the opposite effects according to the context [286,287]. Methylation is also observed at arginine R333, R335, and R337 by protein arginine methyltransferase 5 (PRMT5), influencing p53 transcriptional activity upon DNA damage [288]. The readout of mono-, di-, or trimethylation targeting comes from the deciphering activity of several different classes of readers: Tudor domains, Pro–Trp–Trp–Pro motif (PWWP) domains, and malignant brain tumor (MBT) repeat domains, sharing homologous core regions [289], and plant homeodomain (PHD) finger, a selective reader for dimethylation and trimethylation marks [290].

p53 ubiquitination was reported for the first time in 1993 [291]. In the finely tuned control of p53 protein amounts, a plethora of E3 ligases (i.e., MDM2, PIRH2, COP1, CHIP, and TRIMs) act coordinately with DUBs, such as ubiquitin-specific protease (i.e., USP7/HAUSP) or peptidase (i.e., USP10), to balance the abundance of p53 in a ubiquitin-dependent manner [292,293]. However, in the complexity of p53 modulation, ubiquitination is not confined to be a mere trigger for degradation. Up until today, MDM2 is the major E3 ubiquitin ligase for p53. Since MDM2 could promote both mono- and polyubiquitination on p53, it controls not only its stability but also p53 localization within the cell [294]. Through the modulation of the ubiquitination rate, MDM2 leads to the “low rate” p53 monoubiquitination, thus retaining p53 in the cytosol and promoting further transcriptional-independent roles. On the other hand, MDM2 forms a “high rate” p53 polyubiquitinated form to be rapidly eliminated [295,296,297]. On this side, MDM2 catalyzes ubiquitin binding to all the lysines within the CTD (K370, K372, K373, K381, K382, and K386) [298] targeted for methylation and acetylation. As described above, this competition represents a crosstalk between PTMs acting in p53 functional control. However, in a ubiquitin-independent fashion, MDM2 itself undergoes p53 transcriptional control and is able to inhibit the p53 transcriptional effect by direct binding, thus closing a fine regulatory loop with p53 [299,300,301].

Similar to MDM2, several E3s can ubiquitinate p53 and control its function without triggering any degradation program, as reported for WWP-containing proteins and male-specific lethal 2 (MSL2) involved in p53 cytoplasmic localization [302,303]. Interestingly, E4F1 is an E3 able to ubiquitinate p53 at three lysines (K319, K320, and K321) other than the canonical six cited above, to promote cell cycle arrest [304]. The aberration in p53 E3 ligase functioning could lead to neoplasmic outcomes [305,306].

The same lysine sites interested by ubiquitination can undergo different modifications by several UBL proteins, such as SUMO and NEDD. Unlike ubiquitin, NEDDylation and SUMOylation do not influence p53 stability but control protein localization and functionality [307,308,309].

As in ubiquitination, p53 NEDDylation can be mediated by the activity of MDM2 occurring at three (K370, K372, and K373) of the six mainly ubiquitinated lysines within the C terminus. Interestingly, the ability of MDM2 to promote both ubiquitination and NEDDylation of p53 depends on MDM2 phosphorylation [310]. Moreover, NEDDylation also occurs at p53 lysines involved in non-degradational ubiquitination (K320 and K321) by the F-box protein FBXO11 [311] and E4F1 [304]. This competition to ubiquitination results in the selective inhibition of p53 transcriptional activity, on one hand, and influences nuclear export by preventing p53 monoubiquitination on the other [312].

p53 undergoes conjugation to SUMO proteins upon DNA damage and oxidative stress. Intriguingly, SUMOylation is reported to occur within the CTD at Lys386, usually ubiquitinated but not NEDDylated by MDM2. A number of SUMO E3 ligases promote labeling, including the PIAS family [308] and unless not defined, the precise outcome of SUMO modification on p53 seems to increase its transcriptional activity and in some way enhances p53-dependent senescence [313]. Other researchers have reported that TIP1-mediated p53 SUMOylation prevents p300 binding to the CTD, thus inhibiting transcriptional function [308,314,315]. There is evidence to suggest that SUMOylation of p53 restrains acetylation and decreases affinity to chromatin, thus affecting transcriptional activity [315,316]. As normal, SUMOylation can be reversed by the SUMO protease 1 (SENP1) [317].

Recently, other than SUMO and NEDD, a novel ubiquitin-like protein has been identified. HOPS/TMUB1 (Hepatocyte Odd Protein Shuttling/Trans Membrane Ubiquitin-Like Protein 1, hereafter referred to as HOPS) [318] is able to impact the control of p53 stability by inhibiting its recruitment to the proteasome and lengthening its half-life [319]. Firstly, HOPS has been shown to interact and control the stabilization and the nucleolar localization of the tumor suppressor p19^Arf^, also acting in its functional binding to the onco-suppressor nucleophosmin (NPM) [320]. Given the nature of HOPS interaction to p53 and other targets, it has been referred as HOPSylation, since it acts through the HOPS UBL domain. HOPSylation appears to occur at Lys320 and is reported to be engaged in ubiquitination, NEDDylation and acetylation, and is involved in the management of p53 cellular distribution. Functionally, HOPS interaction to p53 leads to the control of its apoptotic potential upon DNA damaging stresses, impairing p53 localization as a critical event for specific functional outcomes [319,321].

Emerging research shows that p53 can be a substrate for UFMylation. Analogously to other UBLs, the ubiquitin-fold modifier (UFM) is a recently identified ubiquitin-like protein [322,323]. UFMylation occurs at CTD both at lysine sites, and is reported to be interested in other Ub or UBL conjugations (K370 and K373), and two novel ones (K351 and K357). Since p53 UFMylation and other ubiquitin-like modifications compete for the same lysine sites in p53, they act in modulating the stability, by interfering with p53 ubiquitination-mediated proteasomal degradation and possibly in controlling other functional outcomes [324].

## 8. Conclusions

The fine tuning of protein expression and function is a key step in cellular homeostasis and healthy state preservation. Alterations affecting such events including protein biosynthesis and turnover are at the origin of a wide spectrum of pathologies and diseases, including cancer.

The PTMs represent a powerful mechanism to rapidly modulate specific protein amounts in cells, thus controlling their localization, function, and intracellular dislocation. Since they play a critical role in the control of cellular homeostasis and there are negative outcomes for the alteration of proper protein modifications, their role in cancer progression is definitively assessed.

For this purpose, in the past years, more significant evidence has been produced to suggest their use in new therapeutical approaches in the field of oncologic diseases. So far, a number of new molecules (i.e., ubiquitination pathway inhibitors/modulators) have been developed for clinical use and are now available in current medical practice.

## Figures and Tables

**Figure 1 ijms-23-14480-f001:**
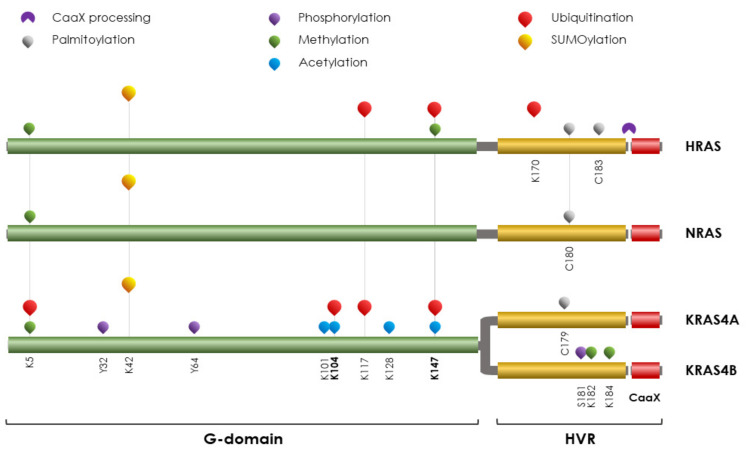
**Schematic representation of RAS isoforms with the indication of the main PTMs**. In humans, three *RAS* genes encode four distinct proteoforms: HRAS, NRAS, and the two splice variants of *KRAS*, KRAS4A, and KRAS4B. The two-thirds at the N terminus of RAS proteins consists of a conserved G-domain (aa 1–165) sharing a high percentage of identity. At the C terminus third, corresponding to the hypervariable region (HVR, aa 166–188, and aa 166–189 for KRAS4B being one aa longer), the proteins undergo a CAAX processing reaction cleavage at the CAAX consensus domain and palmitoylation cycles to direct the membrane trafficking and localization of different entities to drive specific pathway execution. The principal PTMs, including palmitoylation, phosphorylation, acetylation, methylation, ubiquitination, and SUMOylation are reported and labeled as in the figure legends and occur along the whole sequence. The modified aa are highlighted. Due to its critical role in cancer, KRAS4B is better characterized in PTMs.

**Figure 2 ijms-23-14480-f002:**
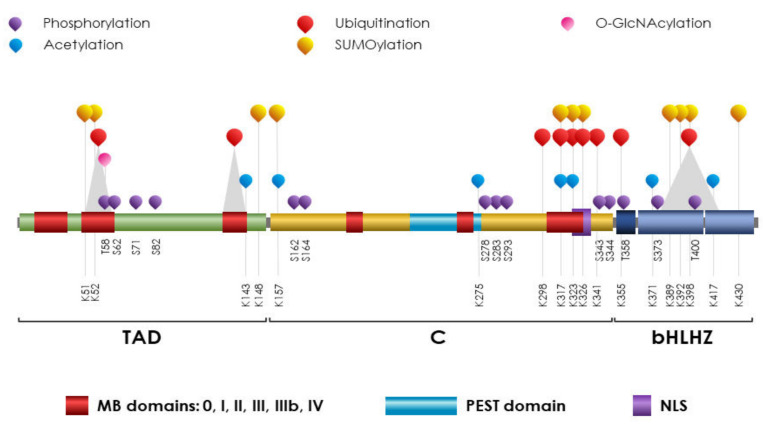
**Schematic representation of c-MYC protein domains with the main PTMs**. Structurally, c-MYC is a 439-amino-acid protein subdivided into three domains. The TAD (aa 1–143) (Trans-Activation Domain) or N-terminal Transactivation Domain (NTD) is responsible for transactivation and trans-repression of target genes. The Central domain (C) surrounding the central area, includes an acidic PEST domain (aa 226–269) and an NLS (Nuclear Localization Site, aa 320–328,). The C-terminal bHLHZip domain (aa 355–435) that includes a basic-region (BR), helix-loop-helix (HLH), and leucine-zipper (LZ) mediates the hetero-dimerization to MAX. Only when in complex with MAX, c-MYC binds DNA sequences at the Enhancer-box (E-box) to exert its transcriptional program. Conserved MYC Boxes (MB) observed in TAD (MB0, I, and II) and the Central domain (MBIII, IIIb, and IV) are reported. The principal PTMs, including phosphorylation, acetylation, O-GlcNAcylation, ubiquitination, and SUMOylation are reported and labeled as in figure legend. Grey triangles indicate ubiquitination areas with no identification of the specific binding site.

**Figure 3 ijms-23-14480-f003:**
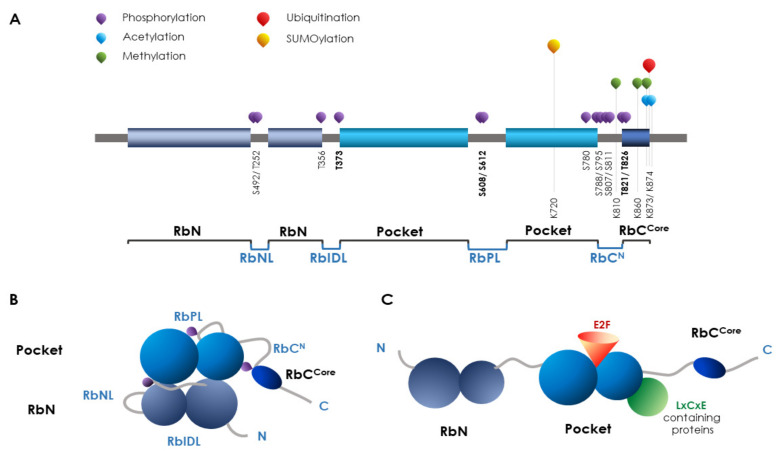
**Schematic representation of RB protein domains with the main PTMs.** (**A**). RB is a 928-amino-acid protein consisting of three large domains. A total of 33% of the whole sequence is made of an intrinsically disordered primary sequence mainly located between the structured domains and is strongly affected by phosphorylation. The N terminal domain (aa 1–379) includes two subdomains (RbNs) linked by a disordered sequence loop (RbNL). The central domain (aa 380–772) is made of two pocket subdomains connected by a pocket linker (RbPL). The pocket domains are required for in vivo interactions with the E2F family members and other partners (i.e., LxCxE containing proteins) are indispensable for Rb growth suppression functions. The core domain (RbC) at the C terminus (aa 773–928) is related to the Rb turnover and protein stability thus affecting tumor outcome. The principal domains are joined by disordered sequences: the interdomain linker (RbIDL) connects the N terminal and central domain; the C terminus includes a disordered RbC^N^ sequence linker. (**B**,**C**). Examples of conformational and functional assets of Rb according to the PTMs pattern. In (**B**), specific phosphorylation events can induce a closed/inactive conformation thus inhibiting protein interactions relevant to different Rb tumor suppressor functions. In (**C**), in dephosphorylated opened/active form, the Rb pocket domains can selectively bind its specific functional partner at the E2Fs site and/or LxCxE binding cleft.

**Figure 4 ijms-23-14480-f004:**
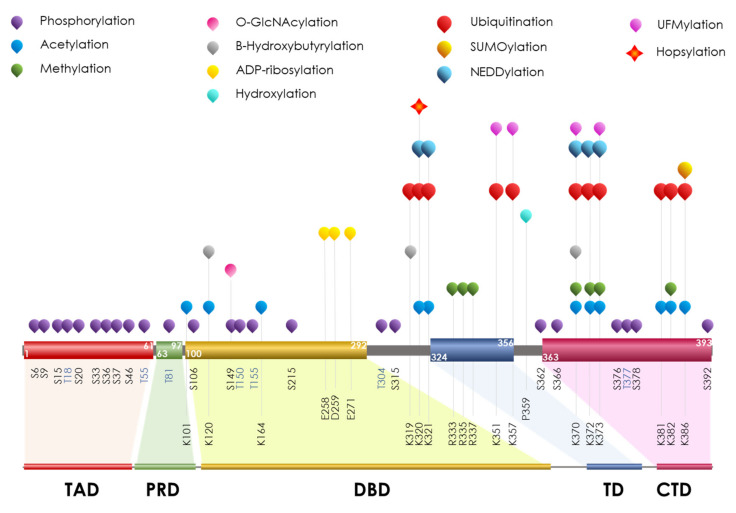
**Schematic representation of p53 protein domains with the main PTMs**. p53 is a 393 aa long protein whose functional domains are: N-terminal Trans Activation Domain (TAD, aa 1–61), Proline-Rich Domain (PRD, aa 64–92), central DNA-Binding Domain (DBD, aa 101–292), Tetramerization Domain (TD, aa 325–356), and an intrinsically disordered C-Terminal Domain (CTD, aa 363–393). Given its regulatory function, the CTD is strongly affected by PTMs. The specific PTMs are functional to the role of the domain involved. All the six p53 domains and relative linker sequences undergo PTMs: phosphorylation, acetylation, methylation, O-GlcNAcylation, ADP-ribosylation, hydroxylation, β-hydroxybutyrylation, ubiquitination, SUMOylation, NEDDylation, UFMylation and Hopsylation are reported and labeled as indicated in the legend. The localization and the modified aas are reported. The higher PTM frequency in the last third of sequence I implies that the picture is not drawn to scale. The real size of each domain is shown at the bottom.

**Table 1 ijms-23-14480-t001:** The four homeostasis knights’ genes.

Gene		Description	Oncogenic Alteration	Related Cancers
** *RAS* **	-*KRAS*	*RAS* Proto-Oncogenes GTPase activity Signal transduction	Mutation Amplification Deep deletion	Pancreatic Colorectal Lung NSCLC
	*-NRAS*		Mutation Amplification Deep deletion	Melanoma AML Endometrial NSCLC
	*-HRAS*		Amplification Deep deletion Mutation	Lung Head and Neck Soft tissue Esophagogastric
** *MYC* **		*MYC* Proto-Oncogene BHLH Transcription Factor	Amplification	Endometrial Ovarian Brest Head and Neck
** *RB* **		Transcriptional Corepressor 1 Tumor suppressor Cell proliferation	Amplification Deep deletion Mutation	Bladder Colorectal Esophagogastric Soft tissue
** *TP53* **		Tumor suppressor growth arrest apoptosis induction cell cycle regulation	Mutation	Ovarian Endometrial NSCLC Esophagogastric

## Data Availability

Not applicable as this review did not report new data.

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
