# Peer review of "The Four Homeostasis Knights: In Balance upon Post-Translational Modifications"

_ijms, 2022, doi:10.3390/ijms232214480_

Round 1

Reviewer 1 Report

It is an interesting review. Please consider the following comments:

1. The ultimate purpose of the review was not described in the introduction section. Certain issues or motive should be specified for the review.

2. Little information has been provided on ADP-ribosyltion, a representative factor in PTM. A separate paragraph on the changes in PARP and ADP-Ribosylation must be included as the main contents. Interplay between ubiquitination, methylation, phosphorylation, SUMOylation, and ADP-ribosylation is an important point in PTMs. It is only briefly introduced in figure legend of Figure 4.

Author Response

We thank the Reviewer1 for considering our paper as an interesting review.

We addressed the reviewer comments as follows:

  1. The ultimate purpose of the review was not described in the introduction section. Certain issues or motive should be specified for the review.

The reviewer is right, since we define the ultimate purpose of the review only in paragraph 3.3, after introducing the main features of PTMs. Surely, we accept the criticism and we added the following sentence at the end of Introduction paragraph:

“According to these observations we will delineate the biological strategies engaged to control the cellular amount of such fundamental protein entities, especially focusing on four of the most important oncogenes/TSGs trough the analysis of the main post-translational modification involved.”

  1. Little information has been provided on ADP-ribosylation, a representative factor in PTM. A separate paragraph on the changes in PARP and ADP-Ribosylation must be included as the main contents. Interplay between ubiquitination, methylation, phosphorylation, SUMOylation, and ADP-ribosylation is an important point in PTMs. It is only briefly introduced in figure legend of Figure 4.

We agree with the reviewer about the relevance of ADP-ribosylation, especially when talking about DNA repair pathways. However, in this paper we focused on the more represented PTMs observed in the 4 cancers related molecules described. Even if ADP- ribosylation is reported for p53, no evidence about RAS, RB and c-MYC were found. Considering the variety of PMTs involved in TSGs and oncogenes control, we needed to choose the most representative ones having impact in all of the four genes described in our paper. Undoubtedly, the role of ADP-ribosylation is relevant and it deserve more attention than what we can do for the final purpose of this paper. This is the reason why we did not consider including ADP-ribosylation in our overview study.

Reviewer 2 Report

Comments:

1. Add a Table to summarize the 4 knights in diseases and PTMs associated.

2. Any info of phosphorylation and acetylation?

3. Any new  or novel PTMs which are discovered recently on those 4 knights?

Author Response

We thank the reviewer for the helpful criticism of and we went to the following conclusions.

  1. Add a Table to summarize the 4 knights in diseases and PTMs associated.

The review is focused on cancer and depicts the impact of PTMs on TSGs and oncogenes activity. A table listing all the diseases related will be an off-topic element. However, we followed the reviewer suggestion, and we considered the insertion of a table summarizing the mainly occurring types of cancers and the oncogenic alteration involved for the four proteins examined. We inserted Table1 at page 5. 

  1. Any info of phosphorylation and acetylation?

For each “knight” we described phosphorylation and acetylation. If the reviewer refers about the presence of boxes illustrating the molecular mechanisms underlying the addiction of phospho- or acetyl groups (as we did for ubiquitination and SUMOylation) we did not provide them for the reason explained in the text at lines162-165)

“The mechanism of modification for the first group is a single or a few steps enzymatic reaction. The process engaged for the second group of modification is quite complex and deserves more details. The PTMs machineries for ubiquitination and SUMOylation are briefly summarized in Box 1 and Box 2.”

  1. Any new or novel PTMs which are discovered recently on those 4 knights

Since TP53 is the protein having the wider variety in PTMs involved in its regulation, we could find new or novel PTMs only in TP53 field (line 729 and 742). Unless so, we cannot exclude that studies around novel PTMs for the 3 other players are in progress and surely intriguing.

Reviewer 3 Report

The manuscript entitled” The four homeostasis knights: in balance upon Post-Translational Modifications” is interesting and scientifically relevant review, the authors have covered the most important tumor suppressors including, retinoblastoma (Rb), p53, phosphatase, etc., however there are some improvements to be made.

1-    Include figure to show Ubiquitinated and de-ubiquitinated p53 functions and pathways.

2-    The authors should include a table to illustrate drugs that are being explored to target Rb, p53 family, and PTEN.

3-    A table showing selected tumor suppressor genes, in the table the authors have to include the following column:

·       Gene

·       Function

·       Associated cancer

·       Others major tumor

Author Response

We thank the Reviewer for considering our manuscript an interesting and scientifically relevant review, and for suggesting some improvements.

We addressed the criticism as follows

  • Include figure to show Ubiquitinated and de-ubiquitinated p53 functions and pathways.

We know how p53 impacts on cellular health and behavior and the field of p53 control by ubiquitination state is certainly a key element for cellular homeostasis. However, the question is so complex and interconnected with other cellular pathways to be difficult to include in a review focused on the whole panorama of PTMs (not only ubiquitination) and extended to four different genes. The introduction of a new figure with the suggested purpose will require also some paragraphs illustrating, unless briefly, the pathways reported. This is the reason why we didn’t analyze so deeply this topic and we preferred to maintain our thread of speech.

2-    The authors should include a table to illustrate drugs that are being explored to target Rb, p53 family, and PTEN.

In this paper we detailed the impact on cancer outcome without considering the pharmacological approach in targeting specific PTMs sites in the analyzed molecules or pathways. The suggested purpose to include the new drugs to target PTM pathways is very interesting, but it surely deserves a distinct and more detailed overview. The review is about 4 factors only and, even if of great interest and impact on carcer outcome, PTEN is not included in this paper topics.

3-    A table showing selected tumor suppressor genes, in the table the authors have to include the following column:

  • Gene
  • Function
  • Associated cancer
  • Others major tumor

We agree with the reviewer about the introduction of a table summarizing the gene analyzed in our manuscript (both oncogenes and TSGs) with minor modification in the columns content. The table shows: genes, description, oncogenic alteration, related cancers (mainly occurring). We inserted Table1 at page 5. 

Round 2

Reviewer 1 Report

All concerns have been well addressed. 

Reviewer 3 Report

I am fine with the revised version